# An Optimized FI-RSV Vaccine Effectively Protects Cotton Rats and BALB/c Mice without Causing Enhanced Respiratory Disease

**DOI:** 10.3390/v14102085

**Published:** 2022-09-20

**Authors:** Min Lin, Wei Zhang, Yi-Fan Yin, Jun-Yu Si, Lu-Jing Zhang, Li Chen, Xue Lin, Ying-Bin Wang, Jun Zhang, Zi-Zheng Zheng, Ning-Shao Xia

**Affiliations:** State Key Laboratory of Molecular Vaccinology and Molecular Diagnostics, National Institute of Diagnostics and Vaccine Development in Infectious Diseases, School of Public Health, Xiamen University, Xiamen 361002, China

**Keywords:** RSV, vaccine, formalin, pre-F, ERD, antibody avidity

## Abstract

Background: Despite considerable efforts toward vaccine development in past decades, no effective vaccines against respiratory syncytial virus (RSV) are available. Recently, we showed that an optimized formalin concentration can preserve prefusion protein (pre-F) on RSV-infected cells and protect mice against RSV infection without causing enhanced respiratory disease (ERD). Here, we sought to further stabilize pre-F on RSV virions by optimizing the production of FI-RSV. Methods: Freshly produced RSV virions were treated with formalin under different concentrations to obtained an opti-FI-RSV vaccine with high pre-F level. Immunogenicity and safety of opti-FI-RSV were evaluated in Balb/c mice and cotton rats. Results: Using 0.0156–0.1778% formalin, we successfully preserved pre-F on virions. This opti-FI-RSV exhibited improved immunogenicity and efficacy without causing ERD. Surprisingly, opti-FI-RSV, with a pre-F-dominant immunogen, still caused ERD after immunization with a suboptimal dose or when the neutralizing antibody titers declined. ERD was avoided by coadministering opti-FI-RSV with CpG + MPLA adjuvant, which subsequently induced a Th1-biasing immune response and, more importantly, significantly improved antibody avidity. Conclusions: Our study provides a new method to obtain a novel FI-RSV vaccine with a high pre-F level and may provide a reference for developing other inactivated vaccines. Our findings also emphasize that appropriate adjuvants are critical for nonreplicating vaccines.

## 1. Introduction

Respiratory syncytial virus (RSV) is the leading viral pathogen correlated with acute lower respiratory infections (ALRIs) among children under 5 years of age [1,2] and the dominant cause of severe pneumonia requiring hospitalization for children aged 1–59 months [3]. Almost all children younger than 2 have experienced RSV infection, and repeated infections can occur due to low natural immunity [4,5]. In older people, the RSV disease burden is as high as that of influenza A infection [6]. Despite the high disease burden, a vaccine for RSV is currently not available, although numerous vaccines are undergoing clinical trials or preclinical studies [7].

Candidate platforms for vaccines are categorised into five major types: recombinant viral vectors, inactivated viruses, nucleic acid-based vaccines, protein subunit and live attenuated virus. Inactivated vaccines are safe and effective since they cannot replicate at all in an immunised individual or there is no risk of reversion to a wild-type form which is capable of causing diseases. However, in the 1960s, a formaldehyde-inactivated RSV (FI-RSV) candidate vaccine caused enhanced respiratory disease (ERD) during phase II clinical trials without protecting RSV-naïve children [8]. Research over the past decades has shown that FI-RSV induces poor neutralization of antibodies and primes a Th2-biasing response, which subsequently causes enhanced illness [9,10]. The prefusion (pre-F) conformation of RSV glycoprotein is currently the most attractive target for vaccine development since it has been proven to have the capacity to elicit neutralizing antibodies in various vaccine platforms, such as subunit vaccines [11,12], gene-based vaccines [13,14], virus-like particle vaccines [15,16], and recombinant adenovirus vaccines [17]. However, studies have provided evidence that pre-F does not exist on FI-RSV virions, possibly due to loss during production [18]. Formaldehyde has a well-established role as an inactivating agent in the preparation of vaccines for diseases such as diphtheria, tetanus [19], hepatitis A [20] and polio [21]. Despite being widely used, formaldehyde fixation can cause epitope alterations that result in a loss of immunoreactivity [22,23]. We have reported that an optimal concentration of formaldehyde is important for stabilization of the pre-F on RSV-infected cells and that an optimally fixed (“opti-fixed”) immunogen protects mice well against RSV infection without causing ERD [24].

RSV-infected cells have many cell contents that may affect their further use as vaccine candidates [24]. In the present study, we investigated whether an optimal concentration of formaldehyde could preserve pre-F on RSV virions, similar to the case on cell surfaces. We confirmed our observations made on the cell surfaces that compared with FI-RSV, opti-FI-RSV exhibited good pre-F preservation with improvements in thermal stability, immunogenicity, and efficacy without causing ERD in both cotton rats and BALB/c mice. Surprisingly, we also found that despite the high immunogenicity induced by the pre-F-abundant opti-FI-RSV, ERD still occurred upon immunization with a Th2-biasing adjuvant after the decline in neutralizing antibody titers. However, a Th1-biasing adjuvant with the capacity to improve antibody avidity may avoid the risk of ERD. To our knowledge, we are the first to identify that even a pre-F dominant vaccine still carries a risk of ERD after neutralizing the decline of antibody levels, especially when a Th2-biasing adjuvant is used.

## 2. Materials and Methods

### 2.1. Cells, Virus and Animals

HEp-2 (ATCC CCL-23) and Vero (ATCC CCL-81) cells were cultured in Dulbecco’s modified Eagle’s medium (DMEM) containing 10% fetal bovine serum (FBS) with 100 U/mL penicillin–streptomycin (Gibco). RSV strain A2 (ATCC) and RSV-A2-mKate (pSynkRSVA2 D46F, kindly donated by Dr. Barney S. Graham, VRC, NIH; this virus is a recombinant virus based on RSV strain A2 with insertion of a fluorescent reporter before the NS1 protein) were produced in HEp-2 cells. Six- to eight-week-old female BALB/c mice (Shanghai SLAC Laboratory Animal Co., Ltd., Shanghai, China) were maintained under pathogen-free conditions until use. Female cotton rats were bred and housed. All of the animal experiments performed in this study were approved by the Institutional Animal Care and Use Committee and Laboratory Animal Management Ethics Committee of Xiamen University (ethics approval number: XMULAC20200198).

### 2.2. Antibodies and Proteins

Plasmids of antibodies, including AM22, D25, MOTA, MPE8, 101F and 12A12, were kindly donated by Jason S. McLellan (Department of Molecular Biosciences, College of Natural Sciences, The University of Texas at Austin, TX, USA) and expressed using FreeStyleTM 293-F cells (Gibco) as previous described [24].

The soluble post-F protein was constructed as described by McLellan et al. [25], and the soluble pre-F protein SC-TM was constructed as described by Anders Krarup et al. [12]. The soluble pre-F and post-F proteins were expressed in FreeStyleTM 293-F cells (Gibco), and purified by a two-step protocol applying a cation exchange chromatography followed by SEC. The eluate was concentrated and the protein was further purified on a Superdex200 column (GE Healthcare). A reduced SDS–PAGE analysis was used to determine purity of the final protein preparation. The identity of the prefusion conformation and postfusion conformation was verified using ELISA with RSV antibody panel (Appendix A).

### 2.3. Vaccine Preparation

Viruses with high levels of pre-F were produced. Briefly, Vero cells were infected with RSV A2 at an MOI of 0.3, and 63 h post-infection, the cells were scraped and centrifuged at 1500 g for 20 min at 4 °C. The supernatants were collected and titered before being treated with different concentrations of formaldehyde (Sigma-Aldrich, St. Louis, MO, USA) and then incubated at room temperature for 12 h. After dialysis with 550 mM NaCl, the supernatant was preserved at −80 °C until use, as it has been reported that exposure to low-molarity buffer might trigger the pre-F transition to post-F [26]. Before immunization, Alhydrogel (InvivoGen), MPLA (InvivoGen), or CpG (InvivoGen) was mixed with opti-FI-RSV.

FI-RSV was prepared according to the protocol for “Lot-100” with some modifications [27]. The clarified supernatant was filtered through a 5 μM filter (Millipore SM Membrane) and then incubated for 72 h with formaldehyde (1:4000) at 37 ${}^{\circ }$C. The fluid was centrifuged at 50,000 g at 4 °C for 1 h. Subsequently, the pellet was resuspended to achieve 25-fold concentration. Further 4-fold concentration was achieved by precipitation with Alhydrogel at 4 °C for 1 h and resuspension of the pellet in one-fourth of the original volume. The clarified supernatant of mock-infected Vero cells was treated in the same way and designated FI-Mock.

The complete inactivation of virus preparations was tested through passaging the supernatants on HEp-2 cells for multiple passages as previously described [28]. The levels of residual host cell DNA for opti-FI-RSV, FI-Mock, and FI-RSV were determined with a sensitive and specific real-time PCR method as previously described in [29]. Results have been shown in Appendix A.

### 2.4. Thermal Stability

Freshly produced viruses were aliquoted and pooled in 15 mL conical tubes. Some of the tubes were treated with formaldehyde at room temperature, while the untreated tubes were placed under the same conditions but without formaldehyde. Then, the viruses in both types of tubes were dialyzed with 550 mM NaCl as mentioned above. At 18 h post-dialysis, 1.5 mL from each tube was transferred to replicate tubes and incubated at either 4 °C or 37 °C. Quantification of the relative pre-F level was performed at designated time points.

### 2.5. Biotin–Streptavidin-Based Sandwich ELISA

H4E3 was chosen as a capture antibody according to our previous work. This antibody can bind RSV well without affecting the binding of other known epitopes. Secondary antibodies, such as AM22, D25, MOTA, MPE8, 101F, and 12A12, were biotinylated using EZ-Link Sulfo NHS-LC-biotin (Thermo Fisher Scientific, Waltham, MA, USA) according to the manufacturer’s protocol. Ninety-six-well ELISA plates were coated with H4E3 in phosphate buffer (pH 7.4) and incubated overnight at 4 °C. The plates were washed three times with PBS-Tween (PBST, 0.05% Tween 20 in PBS), and then 5% BSA (in PBS) was added for blocking. The plate was incubated at 37 °C for 2 h. To better quantify the F protein on virions or vaccines, SC-TM was chosen as a standard. The samples to be tested were then serially diluted 4-fold. The standard was diluted to a concentration of 2 μg/mL. Biotinylated antibodies were prepared by diluting the antibodies to 1 μg/mL, mixing samples or standards with the diluted biotinylated antibody at a 1:1 ratio, incubating the mixtures at 37 °C for 1 h, transferring the mixtures to a blocked plate, and incubating the mixtures for another 1 h at 37 °C. The plates were washed six times with PBST. Then, poly-HRP streptavidin (Thermo Fisher) was added at a 1:20,000 dilution, and the plates were incubated for 1 h at 37 °C. The plates were washed six times with PBST and developed with TMB at 37 °C for 10 min. The OD450 (reference, OD620) was measured on a microplate reader (TECAN, Männedorf, Switzerland).

A standard curve was obtained by plotting the mean absorbance values against the logarithm-transformed SC-TM concentrations and fitted to a four-parameter logistic equation using GraphPad Prism. We serially diluted the sample and quantified it when the optical density (OD) = 2 since it was in the linear region. The concentration of the tested sample was calculated based on the equation
(1)$$Y=\frac{2}{10^{a}}\times 10^{b}\;$$
where a is the *X* value when the OD equals 2 for the standard and *b* is the *X* value when the OD equals 2 for the sample. The ratio of the concentration of AM22 to the concentration of MOTA was calculated to determine the pre-F level normalized to the total F level. For immunization, all groups were immunized with an equal total F level.

### 2.6. Animal Vaccination and RSV Challenge

Female cotton rats (6–8 weeks old) were randomly grouped and vaccinated intramuscularly in the thigh of the left leg on days 0 and 21 with FI-Mock, FI-RSV or opti-FI-RSV, the immunization dose per animal was 1 μg of total F quantified by Biotin–streptavidin-based sandwich ELISA. Serum samples were obtained on day 14 post-boost for determination of neutralizing antibody titers or conformation-specific titers. To assess efficacy, cotton rats were challenged by the intranasal route on day 28 post-boost with $1\times 10^{6}$ plaque-forming units (PFU) of RSV strain A2. On day 5 post-challenge, the right lung was homogenized, and the virus titer was determined by plaque assay as previously described [24]. The left lungs were fixed with 10% neutral formaldehyde and paraffin-embedded.

The slides were then scored by a pathologist who was blinded to the groups on a scale of 0 to 4 based on the severity of perivascular mononuclear inflammatory cell infiltration, peribronchial mononuclear inflammatory cell infiltration, interstitial pneumonitis, and alveolitis. Slides were scored blindly on a 0–4 severity scale as previously described [30].

Female BALB/c mice (6–8 weeks old) were vaccinated intramuscularly in the same manner as the cotton rats with opti-FI-RSV in the presence of 50 μg of Alhydrogel, 2 μg of CpG + 1 μg of MPLA, or phosphate-buffered saline. To evaluate the antibody isotype and affinity, serum samples were collected on day 14 post-boost and further collected monthly until 9 months post-boost for determination of the neutralizing antibody titers. At 28 days post-boost, BALB/c mice were challenged with $2\times 10^{6}$ PFU of RSV strain A2 by the intranasal route. On day 5 post challenge, the lung tissues were removed and processed in the same way as the cotton rat tissues for determination of virus titers and lung pathology.

### 2.7. IgG Subclass and Avidity Immunoassays

Serum IgG isotype analysis was performed as previously reported [24]. Briefly, 96-well ELISA plates were coated with SC-TM in phosphate buffer (100 ng/well) and incubated overnight at 4 °C.The plates were washed three times with PBS-Tween (PBST, 0.05% Tween 20 in PBS), and then 5% BSA (in PBS) was added for blocking. The sera were then serially diluted 5-fold, transferred to an antigen-coated plate, and incubated at 37 °C for one hour. The plates were washed six times with PBST, and then HRP-conjugated goat anti-mouse IgG, IgG1, and IgG2a antibodies were added at 1:5000 dilutions for the different IgG sub-classes. The serum endpoint titer is expressed as the highest dilution of the serum giving an absorbance reading value greater than 0.3.

Avidity was determined by modifying the above assay to include a 10-minute wash with 7 M urea. The serum was diluted 1:1000 in this assay. The percent avidity was calculated as previously described [31,32].

### 2.8. RSV F Competitive ELISA

The levels of serum antibodies specific for pre-F and post-F were determined with a competitive ELISA as previously described with minor modifications [33]. Briefly, 96-well ELISA plates were coated with pre-F or post-F overnight at 4 °C. Immunized cotton rat serum was serially diluted 4-fold and mixed (1:1 ratio) with 20 μg/mL pre-F or post-F for 1 h at 37 °C. After incubation, 100 μL of the mixture was transferred to either a pre-F or post-F coated plate and incubated at 37 °C for another 1 h. HRP-conjugated cotton rat IgG (Immunology Consultants Laboratory) was added, followed by TMB substrate. The serum endpoint titers were determined as described above.

### 2.9. Neutralization Assay

The levels of neutralizing antibodies against RSV strain A2 were measured with Hep-2 cells and RSV-A2-mKate (pSynkRSV A2 D46F) [34]. In short, 10 μL of immunized serum or monoclonal antibody (mAb; 1 μg/μL) was added to 90 μL of DMEM, serially diluted 4-fold, mixed with 75 μL of RSV-A2-mKate and incubated at 37 °C for 1 h. During the incubation, 96-well microplates were seeded with HEp-2 cells at a density of 30,000 cells per well. The mixtures were transferred to the cell plate for incubation. The fluorescence intensity was captured with a SpectraMax Paradigm Multi-Mode Microplate Reader (Molecular Devices, LLC, San Jose, CA, USA) at 588 nm excitation and 633 nm emission after 24 h. The half-maximal inhibitory concentration (IC50) values were computed with GraphPad Prism version 8.00 (GraphPad Software, San Diego, CA, USA)

### 2.10. Plaque Assay

The titer of virus stock or lung homogenate was determined as described previously [24]. Briefly, virus stocks or lung homogenate were serially 10-fold diluted with MEM medium. Then 50 μL of the diluted samples from the dilution of $1\times 10^{3}$ to $1\times 10^{7}$ were added to a monolayer of HEp-2 cells ($2\times 10^{5}$ per well, 12-well plate) for one hour at 25 °C, and then transferred into the 37 °C incubator with 5% CO_2_. Four days later, plaques were stained with hematoxylin and eosin (H&E) (Sigma-Aldrich) and counted.

### 2.11. Statistical Analysis

Statistical assumptions were assessed prior to analysis. If statistical assumptions were not violated (in the case of three or more groups), an analysis of variance (ANOVA) followed by appropriate multiplicity-adjusted tests (e.g., Tukey’s or Dunn’s test) was performed. If the assumptions were violated, a Kruskal-Wallis test (nonparametric) followed by an appropriate multiplicity-adjusted test was performed.

## 3. Results

### 3.1. Pre-F on RSV A2 Virions Is Rapidly Lost within 3 Days

Metastable pre-F undergoes a dynamic transition to form a stable postfusion bundle that facilitates viral and host membrane fusion [24]. Considering that both pre-F and post-F are present on RSV virions, we first tried to evaluate the relative amount of pre-F antigen on the virions using an ELISA-based approach to compare the antibody binding of AM22 (which binds to site Ø on pre-F) and motavizumab (which binds to site II on both pre- and post-F) [11,35]. We found very low levels of pre-F on RSV virions in prepared stocks, especially under ultrasonic conditions (Appendix A). To obtain RSV stocks with high levels of pre-F, we optimized the production procedures for the virus. We used RSV A2 at different multiplicities of infection (MOI) to infect Vero cells and collected the virus at different time points after infection. When the MOI was adjusted to 0.1–0.5, and the infection time was shortened to 60–63 h, we successfully obtained RSV stocks with a high pre-F level (Figure 1a). We then tested the thermostability of pre-F on RSV at 4 °C and found that the relative level of pre-F declined rapidly within 1 day and was almost absent on day 3 (Figure 1b).

### 3.2. Pre-F-Specific Epitopes Are Preserved Well on RSV A2 Virions by an Optimized Formaldehyde Concentration with Improved Thermostability

It has been reported that pre-F is absent from the surface of FI-RSV, which subsequently induced high titers of binding antibody with weak neutralizing and fusion-inhibitory activity. These antibodies in the context of large antigen load led to immune complex deposition and complement activation in airways upon subsequent RSV infection and thus induced ERD [18]. We therefore evaluated whether an optimized formaldehyde concentration can preserve pre-F on virions as it does on cell surfaces [24]. Freshly produced virus stocks were treated with different concentrations of formaldehyde, and the original virus stocks were treated in the same manner except that no formaldehyde was added. As previously reported [18], we found that the relative binding of pre-F decreased rapidly under treatment with lower concentrations of formaldehyde (0.0000–0.0104%) (Figure 2a), while both the pre-F binding and the total F binding were nearly lost under treatment with higher concentrations (0.2667–2.015%) (Figure 2a). Surprisingly, within specific concentrations (0.0156–0.1778%) (Figure 2a), we found that both the pre-F and the total F were preserved and that the pre-F level was significantly higher than that of untreated virus stocks. To explore whether formaldehyde fixation improved the stability of pre-F on virions, thermostability was later evaluated, and 0.0527% was selected as the preferred concentration for subsequent experiments. We found that RSV A2 virions treated with the optimized formaldehyde concentration were more thermostable than untreated RSV virions when preserved at 4 °C or 37 °C with a prolonged incubation duration (Figure 2b,c). These data demonstrate that an optimized formaldehyde concentration can not only preserve the pre-F on virions but also improve thermostability.

### 3.3. Opti-FI-RSV Exerts Immunogenic and Protective Effects without Causing ERD in Cotton Rats

A stable pre-F molecule with excellent immunogenicity has been validated in mice and macaques [11]. We speculated that a formaldehyde-treated vaccine that preserves the pre-F antigen could induce higher neutralizing antibody titers and protect animals from viral challenge better than FI-RSV. The antigenic properties and the infectivity of both FI-RSV and opti-FI-RSV were determined before immunization. Compared with FI-RSV, opti-FI-RSV showed a higher binding profile for pre-F specific antibodies (AM22 and D25, both target site Ø on pre-F) (Figure 3a,b). Both FI-RSV and opti-FI-RSV show no infectivity in HEp-2 cells (Appendix A). The immunogenicity of opti-FI-RSV was next evaluated in cotton rats, an accepted animal model for evaluation of ERD. At 2 weeks after two-dose vaccination, sera were collected, and competitive ELISA was performed to evaluate the level of antibody response to pre-F and post-F [11]. After incubation with post-F, the binding capacity of pre-F was almost completely lost in sera of FI-RSV-vaccinated rats, while it was retained at a higher level in sera of opti-FI-RSV-vaccinated rats (Figure 3c), indicating that pre-F was preserved well in opti-FI-RSV even in vivo. As we speculated, opti-FI-RSV with an intact F antigen induced significantly increased levels of serum neutralizing antibody titers above the protective threshold [36] (Figure 3e). Cotton rats were challenged intranasally with RSV A2 at week 7, and the viral loads in the lungs were measured 5 days later. The viral loads were reduced to non-detectable levels in the lungs of all animals immunized with opti-FI-RSV (Figure 3f). To determine the impact of opti-FI-RSV vaccination on the pulmonary histopathology of cotton rats, tissue sections were examined after staining with hematoxylin and eosin (H&E) (Figure 4a–d). FI-RSV immunization resulted in significantly higher vascular and interstitial inflammation scores than FI-Mock and opti-FI-RSV immunization groups. Alveolitis was also more severe in the FI-RSV immunized group, although there was no statistical difference. Both FI-Mock and opti-FI-RSV immunization resulted in mild pathology after RSV infection. Taken together, these results demonstrate that a pre-F-preserved, formaldehyde-treated vaccine can induce increased neutralizing antibody titers and protect animals against RSV infection without causing ERD.

### 3.4. Immunization with opti-FI-RSV at Suboptimal Doses Primes ERD in Cotton Rats

Immunization with low doses of recombinant RSV F protein in both the pre-F and post-F conformations induces ERD in cotton rats and BALB/c mice [36,37]. Since recombinant proteins and opti-FI-RSV are both non-replicating vaccines, we next evaluated whether ERD occurs upon immunization with suboptimal doses of opti-FI-RSV. Cotton rats were immunized with opti-FI-RSV at various doses (0.00016–1.6 μg, quantified by sandwich ELISA). After two doses of immunization, blood samples were collected to determine the neutralizing antibody levels prior to the challenge. Five days after the challenge, the lungs were harvested to determine the virus titers and histopathology scores. Neutralizing activity against RSV A2 was strongly induced at a dose equal to the FI-RSV immunization dose, as described above, and decreased along with a decrease in dose (Figure 5a).

RSV titers were determined in the lungs, which revealed that vaccination with opti-FI-RSV conferred almost complete protection above 0.0016 μg, while only partial protection was provided at 0.00016 μg (Figure 5b). Histopathology analysis was performed to determine if opti-FI-RSV primed ERD when immunization was performed at suboptimal doses (Figure 5c). The lungs of animals immunized with a dose of 1.6 μg or with lower doses of 0.16 μg and 0.016 μg showed no or minimal lung inflammation, similar to previous observations (Figure 4) and consistent with the results of the neutralizing assay. However, animals immunized with 0.0016 μg showed severe lung disease including alveolitis and peribronchiolitis despite complete protection. Those immunized with the lowest dose (0.00016 μg) also showed minimal lung inflammation, but the neutralizing antibody titers were nearly undetectable. It appeared that at the lowest antigen dose, the immune response induced was too mild to contribute to histopathology after RSV challenge [36]. Our results confirm previous findings that inflammation induced by low doses of non-replicating RSV vaccines leads to histopathology even if the vaccines are in a pre-F conformation [36].

### 3.5. A Combination Adjuvant Modulates Immune Responses to Prevent ERD, Possibly by Promoting Antibody Affinity Maturation

The FI-RSV vaccine primes ERD for two main reasons: because RSV challenge induces a Th2 polarization immune response in the lungs and because non-neutralizing antibodies without affinity maturation against RSV are produced in seronegative individuals [38]. Previous studies have proven that rats immunized with low-dose F antigen combined with the Th1 adjuvant GLA-SE (a Toll-like receptor 4 [TLR-4] agonist) still exhibit ERD [32]. In our previous work, the TLR-4 agonist MPLA was combined with opti-FI-RSV and induced a Th1 immune response, while the antibody avidity was as low as that of adjuvant alum hydroxide. However, MPLA combined with the adjuvant CpG, a TLR-9 agonist, induced high-avidity antibody production (Appendix A). We thus investigated whether antibody affinity plays a role in ERD upon immunization with a suboptimal dose of RSV antigen. Cotton rats were immunized with opti-FI-RSV plus Alum or CpG +MPLA at a low dose as described above. After two doses of immunization, sera were collected, and F-specific isotype antibodies were measured. As expected, the CpG + MPLA group showed an elevated titer of total IgG and IgG2a, indicating that a Th1 immune response was induced (Figure 7a).

The neutralizing antibody titers were below the protective threshold in both groups, although the titers in the CpG + MPLA group were higher than those in the Alum group (Figure 7b). In addition, CpG + MPLA with RSV antigen at a suboptimal dose still helped to induce the production of high-avidity pre-F specific antibodies (Figure 7c). We next challenged cotton rats after two doses of immunization and determined RSV titers in the lungs 5 days after infection. Immunization with either CpG + MPLA or Alum significantly reduced the virus titers (Figure 7d). Histopathology analysis was then performed. As expected, cotton rats immunized with the Al adjuvant showed ERD associated with elevated peribronchiolitis and alveolitis. Nevertheless, a suboptimal dose of opti-FI-RSV plus CpG + MPLA effectively prevented histopathological inflammation in the peribronchiolar and alveolar regions (Figure 7e–h). Our results showed that, as opposed to Al, a specific combination of adjuvants can provide effective protection and prevent ERD in cotton rats, even if the antigen dose is non-optimal.

### 3.6. Opti-FI-RSV plus CpG + MPLA Adjuvant Still Provides Protection after the Neutralizing Antibody Levels Decline and Prevents ERD

Given that immunization of cotton rats with a suboptimal dose of RSV antigen induced a low level of neutralizing antibody production, which subsequently primed ERD, we next combined opti-FI-RSV with either Alum or CpG + MPLA and evaluated whether ERD occurred when RSV neutralizing antibodies decreased to a certain low level (Approximately equal to the neutralization titer at a suboptimal dose of 0.0016 ug immunization). BALB/c mice were chosen for this experiment. After two doses of immunization, sera were collected at the designated time points. To confirm that CpG + MPLA induced a Th1-biasing response, F-specific isotype antibodies were first measured. As expected, IgG2a antibody titers were significantly higher with CpG + MPLA than with Alum; the total IgG antibody titers were also elevated, although not significantly (p=0.1095) (Figure 8a).

Additionally, CpG + MPLA helped to produce high-avidity pre-F specific antibodies (Figure 8b). The neutralizing antibody titers reached the highest levels at 30 days post-boost and remained at a hign level until 240 days, although gradual declines were observed in both groups (Figure 8c). The first challenge was performed at 30 days post-boost, as described in cotton rats. Immunization with opti-FI-RSV with either Alum or CpG + MPLA completely protected BALB/c mice from RSV challenge when the neutralizing antibody titers were high without causing ERD (Figure 8d,e). We then challenged the BALB/c mice at 270 days post-boost, as the antibody titers were decresed to a certain low level. Surprisingly, CpG + MPLA still provided complete protection even though the neutralizing antibody titers had declined, and no ERD was observed in this group (Figure 6g,h). However, 2 of 5 BALB/c mice showed breakthrough infection in the Alum immunization group, with severe lung inflammation, especially alveolitis. These data demonstrate that when the neutralizing antibody titers have declined to a certain low level, an inappropriate adjuvant might cause ERD in RSV-naïve animals, while a Th1-biasing adjuvant with the capacity to improve antibody avidity may avoid the risk of ERD.

## 4. Discussion

Despite decades of efforts to develop a safe and effective RSV vaccine, such a vaccine is still not available. The failure of FI-RSV has cast a great shadow over RSV vaccine development, especially with regard to safety concerns. Recapitulation of the FI-RSV-induced ERD in many preclinical models has provided evidence that an effective antibody response (with antibody-neutralizing ability and antibody avidity) and a Th1-biasing immune response may be enough to provide safe and protective immunity even in RSV-naïve individuals [34].

Recent studies on the conformation of the RSV fusion protein have elucidated the important role of the pre-F protein as a candidate vaccine. Antibodies elicited by pre-F show higher neutralizing capacity than those elicited by post-F [11] and determine the magnitude of RSV-neutralizing activity in human serum [39]. Compared with infectious RSV, FI-RSV predominantly presents post-F on the surfaces of its virions, indicating that a low concentration of formalin can alter not only the infectivity of the virus but also the antigenic profile [18]. We showed in a previous study that an optimal concentration of formalin is key to stabilizing pre-F on RSV-infected cells [24]. Here, we optimized the production methods of RSV and obtained RSV stocks with high pre-F levels. We further demonstrated that a proper concentration of formalin can stabilize pre-F on RSV virions with established thermal stability and even in vivo, since immunization of cotton rats with opti-FI-RSV induced the production of a certain percentage of pre-F-specific antibodies. This pre-F-stabilized opti-FI-RSV was immunogenic and highly efficacious in both cotton rats and BALB/c mice.

Formaldehyde is frequently used to produce whole inactivated vaccines, as it can cause protein, ribonucleic acid, or deoxyribonucleic cross-linking, either homogeneously or heterogeneously. In view of the high reactivity of formaldehyde, small changes in incubation time, pH, temperature, and concentration can affect the degree of antigen modification [40,41]. Raie Jadidi et al. found that the process for treating influenza virus with formalin could be optimized with small changes, including changes in the concentration and incubation time [42]. Notably, in the current study, we changed not only the formaldehyde concentration but also the incubation temperature and time correspondingly; specifically, we reduced the incubation temperature to 25 °C and shortened the incubation time to only 24 h. The significant advantages of inactivated vaccines over other vaccines are their safety and simplicity, especially in the context of a newly emerging pathogen without defined mechanisms. However, before the 1980s, inactivated vaccines were developed largely using empirical approaches, and vaccine creation pursued only the inactivation of microorganisms [43]. Recent advances in scientific technologies and in our knowledge of how protective immune responses are induced have enabled us to rationally design novel and safer vaccination strategies [44]. In the case of vaccines depending on induction of neutralizing antibody production to achieve protective effects against viruses such as RSV, an intact protective antigen is of the utmost importance. For such pathogens, optimization of inactivation conditions during vaccine production is essential.

Although it has abundant pre-F on the surface of its virions, opti-FI-RSV still caused ERD in RSV-naïve animals under a non-optimal dose and with the Al adjuvant, which is consistent with previous research [36]. We also found that even after immunization with a prophylactic dose, ERD still occurred when the neutralizing antibody titers declined below the protective threshold. The ERD caused by FI-RSV produced in the 1960s has been ascribed to a Th-2 biasing immune response induced by the Al adjuvant [45]. Michael P McCarthy et al. found that ERD caused by a low dose of RSV antigen still occurred regardless of whether a Th1-biasing adjuvant was used [36]. However, our research showed that a novel combination of adjuvants (CpG and MPLA) could not only induce a Th1-biasing immune response but also promote antibody avidity to avoid the risk of ERD either upon immunization with a suboptimal dose of RSV antigen or when the neutralizing antibody titers declined. A lack of antibody affinity maturation due to poor Toll stimulation leading to enhanced RSV disease has been confirmed by Maria Florencia Delgado et al. [31]. Compared with infectious RSV, inactivated RSV stimulates only TLR4 via the F protein, while infectious RSV stimulates both TLR4 and TLR7 via the F protein and RSV genome, respectively [46]. Differences in the ability to stimulate TLR receptors lead to differences in antibody affinity maturation between infectious virions and inactivated vaccines. Opti-FI-RSV plus only MPLA, a TLR4 agonist, could not induce the production of high-avidity antibodies, while opti-FI-RSV plus MPLA combined with CpG, a TLR9 agonist, did promote the production of high-affinity antibodies. Co-administration of CpG with non-replicated vaccines, such as VLP or subunit-based vaccines, enhances antibody affinity in the contexts of HBV infection [47], influenza [48], and anthrax [49]. Here, we further found that co-administration of MPLA with CpG induces the production of higher-avidity antibodies, thus avoiding the risk of ERD upon immunization with a suboptimal dose of RSV antigen in RSV-naïve animals.

One inherent limitation of our study is that although we optimized the production procedure of RSV virions, a high pre-F RSV stock is still not easy to obtain. The percentage of pre-F depends largely on the virus strain. We used RSV strain A2 here, but other studies have proven that RSV strain A2-line19F exhibits higher relative binding to D25 than strain A2. Reverse genetic approaches should be considered to stably obtain high-pre-F RSV stocks. Another limitation of our research is that the relationship among neutralizing antibody titers, the Th-biasing response, and antibody avidity has not been clearly explained. Abundant neutralizing antibodies protect RSV-naïve individuals from virus infection and avoid the risk of ERD. However, when the neutralizing antibody titer drops to a certain low level, ERD still occurs regardless of whether a Th1 response or a Th2 response is induced. Although antibody avidity plays a critical role, we cannot conclude that high-avidity antibodies alone can avoid the risk of ERD; thus, further efforts should be made to elucidate the underlying mechanism. The third limitation is that only fusion protein has been taken into account in current research, however, glycoproteins are also located on the surface of the viral membrane. We compared the antibody titers against glycoproteins in serum immunized with FI-RSV and Opti-FI-RSV vaccine and did not find significant differences (Appendix A). But more experiments are needed to assess the effect of glycoproteins. There is an increasing awareness that responses in males and females can differ [50,51]. We failed to include an analysis of responses to male animals here, special attention should be paid to such an issue in our further study.

In summary, our research demonstrates a novel method to obtain high-pre-F FI-RSV. Compared with FI-RSV, this candidate vaccine exhibits improved thermal stability, immunogenicity, and efficacy without causing ERD in both cotton rats and BALB/c rats. Co-administration of our opti-FI-RSV with CpG + MPLA induces a Th1-biasing immune response and promotes antibody avidity, thus avoiding the risk of ERD when the RSV antigen is used for immunization at a suboptimal dose. This novel combination of vaccine and adjuvants still provides protection and diminishes ERD risk when the antibody titers fall below the protective threshold.

## 5. Patents

There is a patent (Application No.: WOCN108300705) resulting from the work reported in this manuscript.

## Figures and Tables

**Figure 1 viruses-14-02085-f001:**
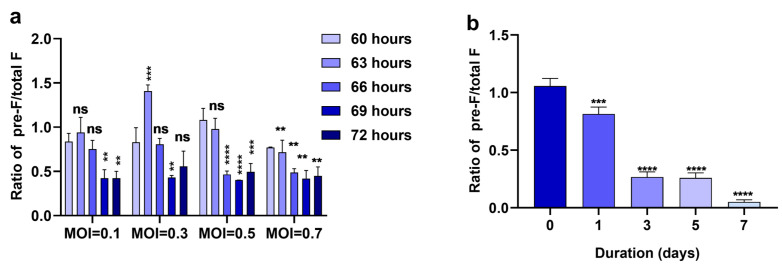
Production of high-pre-F RSV virus stocks and thermal stability assay results. (**a**) Vero cells were infected with RSV A2 at the indicated MOI, and the virus was harvested at the indicated time points. The ratio of the results of sandwich ELISA using AM22, a pre-F-specific antibody, to the results of sandwich ELISA using motavizumab, a total F antibody, is shown. (**b**) The thermal stability of pre-F was tested by incubation with virus at 4 °C, and the ratio of pre-F to total F was calculated by sandwich ELISA at the indicated time points. All bars represent the mean ± SD from three experimental replicates combined and (**a**) the data were analyzed by two-way ANOVA followed by Dunn’s multiple comparisons post hoc test to compare the ratio of 60 h post infection with the ratio of each other time of infection (**b**) the data were analyzed by one-way ANOVA followed by Dunn’s multiple comparisons post hoc test to compare the ratio on day 0 with the ratio on each other day. The *p* values are shown by asterisks (ns, not significant; ** p<0.01; *** p<0.0001; **** p<0.00001).

**Figure 2 viruses-14-02085-f002:**
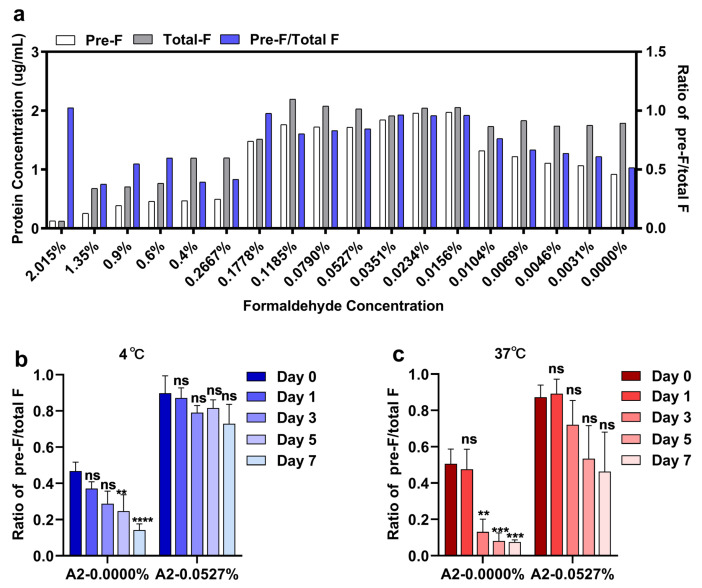
Optimal concentration of formalin to stabilize pre-F in RSV and thermal stability assay results. (**a**) Fresh RSV A2 virions were harvested and treated with the indicated concentrations of formalin at 25 °C for 24 h. Sandwich ELISA was performed to evaluate the concentration of F protein and the relative pre-F level. (**b**,**c**) The pre-F thermal stability was tested by incubation of fixed virus at 4 °C or 37 °C, and the ratio of pre-F to total F was determined by sandwich ELISA at the indicated time points.(**a**) Each bar represents a single measurement. (**b**,**c**) The bars represent the mean ± SD from three experimental replicates combined, and the data were analyzed by two-way ANOVA followed by Dunn’s multiple comparisons post hoc test to compare the ratio on day 0 with the ratio on each other day. The adjusted *p* values are shown by asterisks (ns, not significant; ** p<0.01; *** p<0.0001; **** p<0.00001).

**Figure 3 viruses-14-02085-f003:**
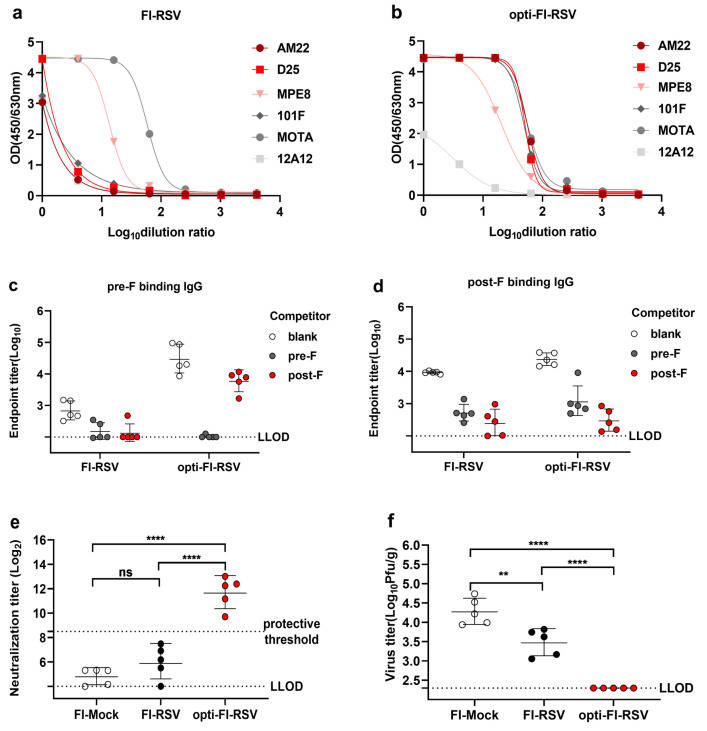
Opti-FI-RSV is immunogenic and protective without causing ERD in cotton rats. Female cotton rats (n = 5 per group) were injected intramuscularly with FI-Mock, FI-RSV or opti-FI-RSV in a prime-boost regimen at week 0 and week 3. Animals were challenged with the RSV A2 strain intranasally at a titer of $1\times 10^{6}$ PFU on day 49. (**a**,**b**) Serial dilutions of the vaccine were incubated with biotinylated pre-F specific antibodies (AM22, D25, AM14) and total F target mAbs (MPE8, 101F, MOTA) or a post-F-specific mAb (12A12), and a secondary mAb conjugated with streptavidin was used to test the binding of the vaccine and the mAb. (**c**,**d**) Pre-F- and post-F-specific antibody levels were determined using a competitive ELISA. (**e**) Serum neutralizing antibody titers were measured using RSV A2 as described in the Materials and Methods. (**f**) RSV virus titers were determined in the right lung homogenate of cotton rats at day 5 after challenge as determined by PFU. The symbols indicate individual rats, the bars/lines indicate the geometric mean ± geometric SD, and the dotted lines indicate the lower limit of detection and the protective threshold. (**e**,**f**) The data were analyzed by one-way ANOVA followed by Tukey’s multiple comparisons post hoc test to analyze the differences between vaccination groups. The *p* values are shown by asterisks (ns = not significant; ** p<0.01; **** p<0.00001).

**Figure 4 viruses-14-02085-f004:**
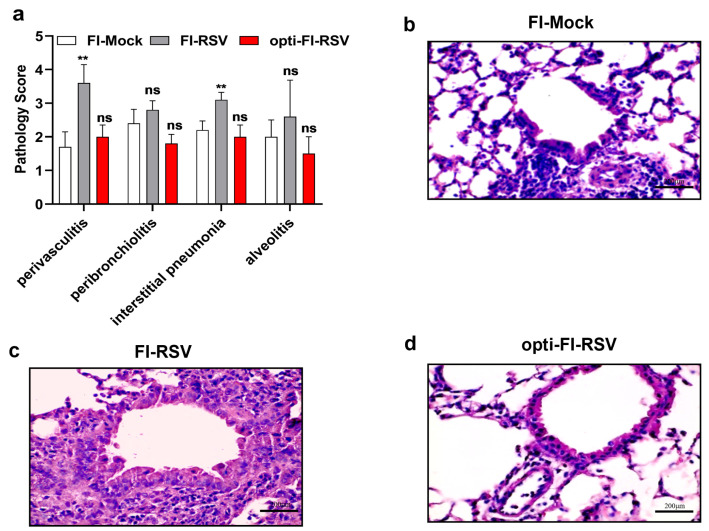
Histopathology following RSV challenge in cotton rats. To evaluate for vaccine attributable enhanced disease post challenge, groups of five cotton rats were inoculated intramuscularly with FI-Mock, FI-RSV, and opti-FI-RSV. All animals were challenged with A2 on day 49 p.i. Left lungs were harvested on day 5 after challenge, and histopathology scoring was performed. Pathology score was calculated as described in the method (**a**). Representative haematoxylin and eosin stains for (**b**) FI-Mock, (**c**) FI-RSV, (**d**) opti-FI-RSV vaccinated rats are shown. Scale bars represent 200 μm. The bars/lines indicate the mean ± SD. (**a**) The data were analyzed by Kruskal-Wallis test (nonparametric) to analyze the differences between vaccination groups. The *p* values are shown by asterisks (ns = not significant; ** p<0.01).

**Figure 5 viruses-14-02085-f005:**
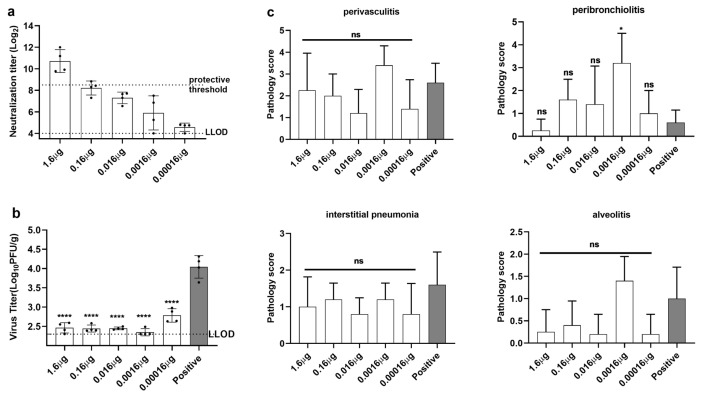
Alum-adjuvanted opti-FI-RSV at a suboptimal dose induces ERD in cotton rats. Female cotton rats (n = 4 per group) were injected intramuscularly with opti-FI-RSV in a prime-boost regimen at week 0 and week 3. Animals were challenged with the RSV A2 strain intranasally at a titer of $1\times 10^{6}$ PFU on day 49. Positive refers to cotton rats that were inoculated intramuscularly with an equal volume of DMEM(mock treatment) (**a**) Serum neutralizing antibody titers were measured using RSV A2 as described in the Materials and Methods. (**b**) RSV virus titers were determined in the right lung homogenate of cotton rats at day 5 after challenge as determined by PFU. (**c**) Left lungs were harvested at day 5 after challenge, and histopathology scoring was performed. Representative H&E staining images are shown in (Figure 6). (**a**,**b**) The symbols indicate individual rats, the bars/lines indicate the geometric mean ± geometric SD, and the dotted lines indicate the lower limit of detection and the protective threshold. (**c**) The bars/lines indicate the mean ± SD. (**b**) The data were analyzed by one-way ANOVA followed by Tukey’s multiple comparisons post hoc test and (**c**) the data were analyzed by Kruskal-Wallis test (nonparametric) to analyze the differences between vaccination groups. The *p* values are shown by asterisks (ns = not significant; * p<0.05; **** p<0.00001).

**Figure 6 viruses-14-02085-f006:**
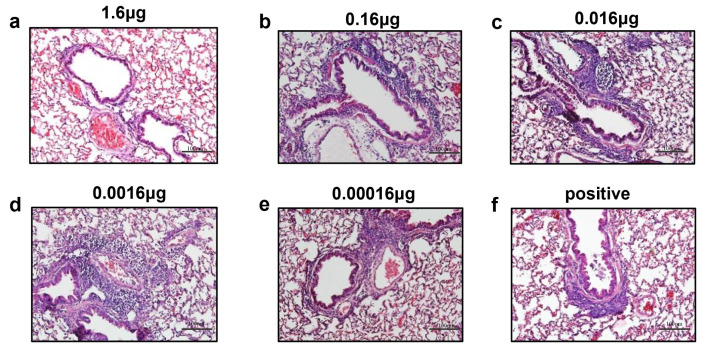
Representative haematoxylin and eosin stains for Alhydrogel-adjuvanted opti-FI-RSV at various doses are shown. (**a**) 1.6 μg, (**b**) 0.16 μg, (**c**) 0.016 μg, (**d**) 0.0016 μg, (**e**) 0.00016 μg, (**f**) positive.

**Figure 7 viruses-14-02085-f007:**
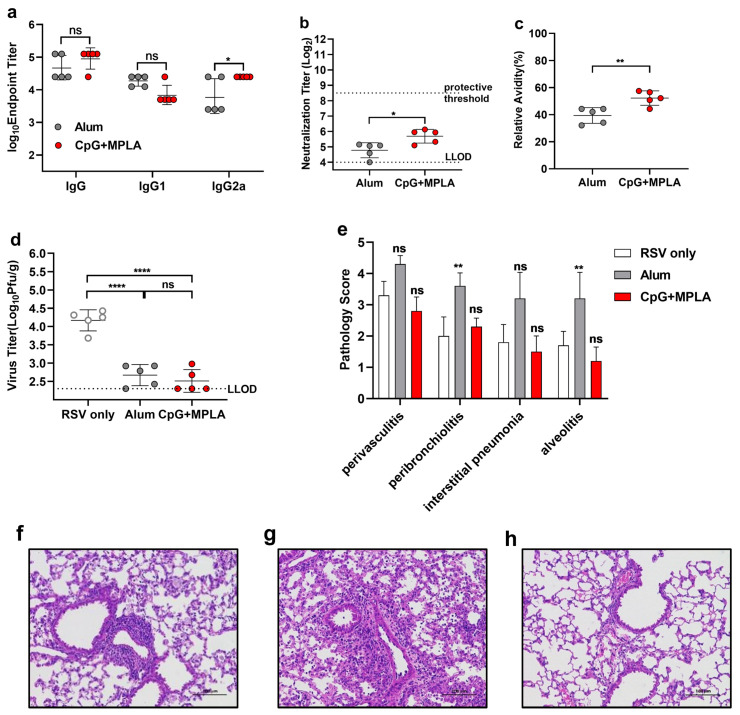
CpG + MPLA adjuvanted opti-FI-RSV avoids the risk of ERD caused by suboptimal immunization. Female cotton rats (n = 5 per group) were injected intramuscularly with opti-FI-RSV combined with a Th2-biasing adjuvant (Alum) or a Th1-biasing adjuvant combination (CpG + MPLA) in a prime-boost regimen at week 0 and week 3. Animals were challenged with the RSV A2 strain intranasally at a titer of $1\times 10^{6}$ PFU on day 49. (**a**) Pre-F specific IgG, isotype IgG1 and IgG2a serum antibodies were determined at 2 weeks after boosting. (**b**) Serum neutralizing antibody titers were measured using RSV A2 as described in the Materials and Methods. (**c**) IgG avidity against pre-F after a 7 M urea wash was determined at 2 weeks after boosting. (**d**) RSV virus titers were determined in the right lung homogenate of cotton rats at day 5 after challenge as determined by PFU. (**e**) Left lungs were harvested at day 5 after challenge, and histopathology scoring was performed. (**f**–**h**) Representative H&E staining images are shown. (**a**,**b**,**d**) The symbols indicate individual rats, the bars/lines indicate the geometric mean ± geometric SD, and the dotted lines indicate the lower limit of detection and the protective threshold. (**c**) The symbols indicate individual rats, and the bars/lines indicate the mean ± SD. (**e**) The bars/lines indicate the mean ± SD. The data were analyzed by (**a**–**c**) t test and, (**d**) one-way ANOVA followed by Tukey’s multiple comparisons post hoc test and (**e**) Kruskal-Wallis test (nonparametric) to analyze the differences between vaccination groups. The adjusted *p* values are shown by asterisks (ns = not significant; * p<0.05; ** p<0.01; **** p<0.0001).

**Figure 8 viruses-14-02085-f008:**
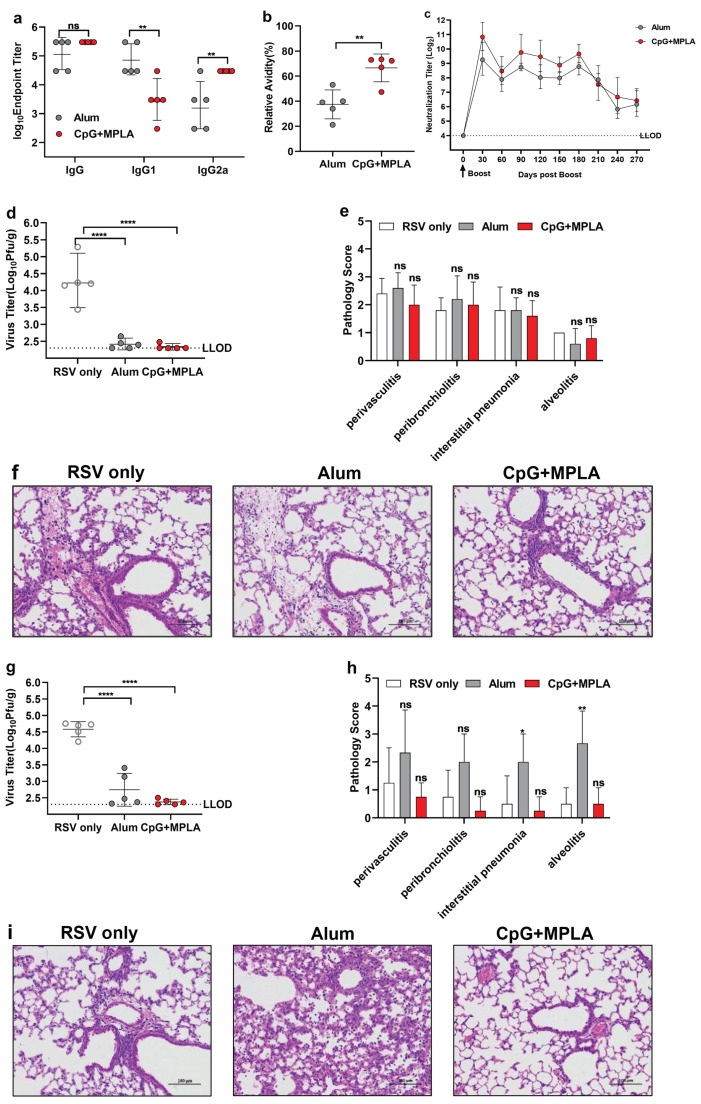
Opti-FI-RSV plus CpG + MPLA adjuvant protects mice from RSV challenge without inducing ERD when neutralizing antibody titers decrease. Female BALB/c mice (n = 5 per group) were injected intramuscularly with opti-FI-RSV combined with a Th2-biasing adjuvant (Alum) or a Th1-biasing adjuvant combination (CpG + MPLA) in a prime-boost regimen at week 0 and week 3. Animals were challenged with the RSV A2 strain intranasally at a titer of $2\times 10^{6}$ PFU on day 49 and day 273. (**a**,**b**) Pre-F-specific IgG, isotype IgG1 and IgG2a serum antibody titers were determined at 2 weeks after boosting, and IgG avidity against pre-F after 7 M urea wash was determined at the same time point. (**c**) Serum neutralizing antibody titers were measured using RSV A2 with the indicated schedule. (**d**,**e**) Lung virus titers and histopathology scores were evaluated at day 5 after the first challenge (**d**,**e**) and second challenge (**g**,**h**). Representative H&E staining images are shown in (**f**,**i**). (**a**–**d**,**g**) The symbols indicate individual mice, the bars/lines indicate the geometric mean ± geometric SD, and the dotted lines indicate the lower limit of detection and the protective threshold. (**g**,**h**) The bars/lines indicate the mean ± SD. The data were analyzed by (**a**,**b**) t test, (**d**,**g**) one-way ANOVA followed by Tukey’s multiple comparisons post hoc test and (**e**,**h**) Kruskal-Wallis test (nonparametric) to analyze the differences between vaccination groups. The adjusted *p* values are shown by asterisks (ns = not significant; * p<0.05; ** p<0.01; **** p<0.0001).

## Data Availability

Not applicable.

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
