# Peer review of "An Optimized FI-RSV Vaccine Effectively Protects Cotton Rats and BALB/c Mice without Causing Enhanced Respiratory Disease"

_viruses, 2022, doi:10.3390/v14102085_

Round 1

Reviewer 1 Report

In the Manuscript titled “An optimized FI-RSV vaccine effectively protects cotton rats and BALB/c mice without causing enhanced respiratory disease” by Min Lin et. al. Authors optimized the methods of producing FI-RSV vaccine with high pre-F levels. The opti-FI-RSV vaccine exhibits good immunogenicity and efficacy without causing ERD in both cotton rats and BALB/c rats. And, the authors further verified CpG+MPLA adjuvants can improve antibody avidity and diminishes ERD risk. Overall, this article is logical, and I believe people who focus on vaccines will learn a lot from the article. However, some issues need to be improved. Introduction to advantages of FI-RSV vaccines should be added. Meanwhile, I think that the authors needs to improve the layout of the figures.

Major

-        L99: please check the title.

-        Figure S1: The texts should be consistent with the figure. Authors should label “a”, “b”, “c” and “d”.

-        Figure 1: The results of fig.1 do not confirm the hypothesis.

-    Figure 3: Figure 3 are too small. The criteria for histopathology scoring should be introduced in the text. In addition, please check that the content conforms to the image. For example, fig.3 does not contain the results of staining with H&E (L246). Are there other assays that can validate the pre-F trimer particles?

-        Figure 5, Figure 6: Do you have any negative control data?

Minor

-        L207-208: Can you explain more about the mechanism of pre-F and ERD?

Reviewer 2 Report

This manuscript is an analysis of immune responses in both mice and cotton rats to immunization with formaldehyde treated respiratory syncytial virus (RSV) virions (FI-RSV) as well as protection of the animals and degree of lung pathology upon RSV challenge.  The authors have defined optimal conditions for formaldehyde treatment of virions as well as production conditions that result in retention of increased levels of the pre-fusion form of the F protein compared to the standard formaldehyde treatment and growth conditions.  Using these conditions to prepare immunogens for animal immunization, the authors characterized the immune responses, protection from RSV challenge, and lung pathologies in the experimental animals.  The authors also characterized effect of suboptimal antibody titers in the animals as well as effects of inclusion of various standard adjuvants.  The authors conclude that optimized preparation of FI-RSV can result in a vaccine candidate that induces high levels of protective antibodies without significant lung pathology (ERD) upon RSV challenge.  They also confirm past results that lower (suboptimal) levels of antibodies result in enhanced pathology and that inclusion of adjuvants decrease this pathology upon RSV challenge.

            The manuscript is clearly written and describes a comprehensive analysis of this RSV vaccine candidate.  There are some concerns with the study that the authors should consider.

1.  The authors have failed to include analysis of responses to male animals.  They only used female animals.  There is an increasing awareness that responses in males and females can differ.

Some journals as well and the US NIH now require inclusion of both sexes in such studies.

2.  There is a lack of statistical analyses of the data presented in some of the figures.  Examples:

Figure 1, panel A

Figure 2, right parts of panels B and C

Figure 3, panels C, D, and G

Figure 4 panels

Figure 5, panel E

Figure 6, panels C, E, H

3.  The virus used as an immunogen was not purified.  As prepared, it likely contains

significant levels of host proteins which could impact immune responses.  Authors should show some data indicating the levels of host protein contamination of their immunogen.

4.  There was no consideration of levels of G protein in the optimized FI-RSV or the role it plays in immune responses.

5.  There is a lack of characterization of the SC-TM F used for targets in ELISA and in quantification of virion proteins.  Was this purified soluble protein?  How was it prepared and validated?  What was the concentration of the protein and was the prefusion conformation confirmed?

6.  There was no clear indication of the dose of F protein used in the experimental animals.  How many micrograms of F protein (and G protein) went into the animals?

7.  Given the difficulties in preparing high titers of RSV and particularly high titers of purified virus, it is not clear how useful this approach will be for preparations of vaccine candidates for clinical trials.

8.  Figure 4:  the authors should explain or at least comment on why the pathology scores go up at 0.0016 ug and then go down at 0.00016 ug.

Other more editorial issues:

9.  Were all animals challenged with RSV by the intranasal route?

10.  Lines 150 and 212:  what pre-F was used?

11.  Line 107:  do the authors mean (24)?

       Line 235:  do the authors mean (35)?

12.  Figure 2:  labeling of panels B and C should include the temperature used.  The figure           legend is not clear.

13.  How were the monoclonal antibodies prepared and used in the study validated?

14 Line 246:  panel f is not histology as stated in the text.

Reviewer 3 Report

Lin et al. describe optimization of the production of formaldehyde-inactivated RSV vaccine by using reduced FA concentration under optimized inactivation temperature and time. The inactivated viruses are compared in regard to their content of the metastable prefusion F protein. The authors conclude that the optimized FA-inactivation protocol not only retains viral antigenic properties but also stabilizes the F protein in the desired immunogenic prefusion conformation. The optimized vaccine was evaluated in cotton rats and mouse model and showed protection against ERD when combined with Th1-driviring immune response adjuvants.

Major points:

-          The authors have presented their data in several well-prepared figures. Nevertheless, for all findings, especially for in vitro methods (Fig. 1 and 2) repetitions of at least three-fold or at least in triplicates are essential. In addition, the calculation of averages and standard deviations needs three independent data points. A presentation of only one data point (data on figure 2a) or two replicates are not strong enough and therefore not valid. Furthermore, in the methods section 2.10. “Statistical analysis”. Did the authors check for the normal distribution of their data before applying the ANOVA test, since ANOVA is only valid for normally distributed data? This applies to all statistical test performed in the manuscript.

-          The authors tested different concentrations of formaldehyde for inactivation of RSV and investigated its impact of the preservation of the pre-F on viral surface. However, the inactivation process was not completely analyzed, since the authors missed the infection of the inactivated material to prove, that the complete virus was inactivated. The complete inactivation of virus preparations is normally tested by a representative amount of material in cell-culture experiments and through passaging the material on cells for multiple passages as published (please check e.g.: Fertey et al. Scientific Reports 2020, PMID: 32732876 or Bayer et al. Vaccine 2018, PMID: 29439869). The uncomplete inactivation might lead to unwanted left infectious viruses in the material for immunization, leading to an enhanced immunity induced by a few active viruses in the Opti-FI-RSV preparations. The authors should include these data.

-          The authors claim in results section 3.1. that MOI of 0.3 and production time of 60-63 hours results in stabilized RSV particles and high content of the prefusion-F. They assume that this might be related to the change of the pH of the medium during virus production. Did they test the pH of the medium to support this claim? In addition, the differences of the ratio pre-F and total F in between the different culture time points (Fig. 1a) are not that high (in contrast to the stability assay in Fig. 1 b). Beside analyzing the ration between pre-F and F total, did the authors check the infectivity of all virus preparations to see correlations for Fig. 1 a and b, respectively? In addition, it is not clearly described, how the ratio of Pre-F to F total can raise over 1. The authors should include the missing data into the manuscript and clearly answer these raised questions.

-          The conservation of the virus antigenicity achieved upon FA-inactivation is not always highly reproducible (see again Bayer et al., Vaccine 2018, PMID: 29439869) and requires more than two experimental replicates to draw valid conclusions.

-          According to the presented figures, the difference in the histopathological changes in the lungs upon vaccination with FI-RSV and opti-FI-RSV are not statistically significant and thus the conclusions regarding any induced ERD should be formulated more carefully in the whole manuscript and also in the title. Please correct the manuscript accordingly.

-          Methods described in section 2.5: explain SC-TM very clear. In addition, the author include data based on measurements of the optical density above 1.5. Please add the standard curve showing linear correlation above OD of 1.5 in the Supplementary information.

Minor points:

-          Line 6: “concentration” should be changed to “concentrations”

-          Line 7: „that“ should be removed

-       Line 10: seems that in this sentence, a preposition “with” is missing: ”opti-FI-RSV, with a pre-F-dominant immunogen”

-          Line 20: delete „of“

-          Line 37: Formaldehyde is referred to as an attenuating agent. This might be inaccurate and should be better described as “inactivating agent”.

-          Line 44: The formulation of the following sentence is unclear: “RSV-infected cells have many cell contents that may affect their further use as vaccine”. What do the authors mean by RSV-infected cells as a vaccine?

-          Line 91 and Line 99: both methods sections 2.4 and 2.5 have the same title “Thermal stability”. Thus, both sections should be either combined in one or a different title should be chosen.

-          Line 104: missing space between “protocol. Ninety-…”

-          In methods section 2.7.: why did the authors skip the blocking step in the ELISA?

-          Line 114: correct “20000” into “20,000”

-          Line 150: include the amount of the F protein used for coating of ELISA plates

-          Line 187: “host membrane fusion24” → the formatting of the reference 24 should be adjusted.

-          Figure 2b and c; please introduce temperature on the figure or clearly in the caption.

-          Line 230-231: the authors should describe the results of the binding profiles in more detail.

-          Results section 3.3: the immunization dose should be included.

-          The histopathological scoring system should be described in more detail.

-          Line 235: the formatting of the reference [35] should be adjusted

-          Line 257: “0.00016 1.6 μg” should be changed to “0.00016-1.6 μg”

-          In general, the authors are encouraged to include the representative histological pictures into the main manuscript instead of presenting these pictures as supplementary information. In figure 4a: the group of 1.6 µg includes only three data point, and in figure 4b, the same group includes four data points. In the figure description, it states that 4 animals per group were tested. This should be explained and adjusted. Please include the exact explanation for the group “Positive”.

-          Line 290: the applied immunization dose should be defined

-          If the authors have analyzed the blood samples (especially for NT-Ab titers, avidity) of the RSV only animals as well, these data should be included in Figure 6.

-          Line 310: use Alum or Alhydrogel instead of Al, also in the Figure caption of Fig. 6. Introduce a full stop after (a)(b) line 10.

-          Introduce the reference for the protective threshold in line 320-321.

-          The explanation in line 330 to 333 is not well-taken, since also the other group showed neutralizing antibody titer below the protection threshold in the absence of break-through infections. Please correct.

-          Please correct the point raised at line 408: “However, when…”, since the authors showed in Fig. 6 neutralizing antibody titers less than the protective threshold associated with an absence of a statistically deference in the induction of ERD in both groups (Al versus CpG+MPLA).

-          Supplement figure 1 “a, b, c, d” are missing in the figure. Please add the described experiments completely in the Methods section (vortex, ultrasonic power, etc.). Correct carefully the text of figures and captions in the Supplement, e.g. Supp 1a: “Nuber of different patch” into “Number of different batch”?

Round 2

Reviewer 2 Report

This manuscript still has problems.  There are still unresolved issues as well as new issues with the revision.

1.  The authors were criticized for using only female animals.  The response to this was to consider the issue in subsequent studies and not repeat these studies with male animals.  They did acknowledge that the literature clearly shows differences in responses of human males and females.  At the least, the authors should add, to the discussion, this issue as a limitation of their study.

2.  The authors did not show levels of host cell contamination in the vaccine candidate as requested by a reviewer.  The authors responded that they will solve purification and scale up protocols with "experienced biopharmaceutical companies".  Most companies like to have a previously determined pathway for these protocols.

3. In the revised manuscript, the authors have failed to update some of the figure legends to correspond to the revised figures.  Examples:

a.  Figure 5 legend:  there is a statement about an H& E image in supplemental figure 4.    Figure 4, neither of them, show such an image.

b.  Figure 7 legend:  again there is a statement about an image in supplemental figure 5.  There is no supplemental figure 5.  Also the legend does not identify panels f, g, and h.

c.  Figure 8 legend:  the authors cite a supplemental figure 6 which does not exist.

Also panels in f and I are not defined.

4.  There are two supplemental figure 4s. 

Reviewer 3 Report

Please refer the first review report. Following major points are not addressed so far.

 The repetitions/replication numbers of experiments are still unclear. For the calculation of averages and standard deviations three independent data points at least are needed. Furthermore, the repetitions are not mentioned in detail for in vitro experiments and also for the in vivo trial (Fig. 5).  

·         I have doubts, that the statistical analyses are correct (normal distribution, as presented).

·         As mentioned, I am convinced that the test of inactivated material needed passaging of the supernatants to ensure the complete inactivation.

·         In addition, some details/data (SC-TM, etc.) are not included into the manuscript.

·         Finally, the definition of the histopathology scores are not clear.
